# Development and Implementation of a Mobile Application for Choosing Empirical Antimicrobial Therapy for Bacteremia, Pneumonia, Urinary Tract Infection, and Skin and Soft Tissue Infection among Hospitalized Patients

**DOI:** 10.3390/antibiotics12010113

**Published:** 2023-01-07

**Authors:** Kanthon Chaloernpoj, Walaiporn Wangchinda, Pornpan Koomanachai, Visanu Thamlikitkul, Pinyo Rattanaumpawan

**Affiliations:** Division of Infectious Diseases and Tropical Medicine, Department of Medicine, Faculty of Medicine Siriraj Hospital, Mahidol University, Bangkok 10700, Thailand

**Keywords:** antimicrobial stewardship, clinical practice guideline, mobile application

## Abstract

Clinical practice guidelines (CPGs) and computerized clinical decision support programs are effective antimicrobial stewardship strategies. The DigitalAMS™, a mobile-based application for choosing empirical antimicrobial therapy under the hospital’s CPGs, was implemented at Siriraj Hospital and evaluated. From January to June 2018, a cross-sectional study was conducted among 401 hospitalized adults who received ≥1 dose of antimicrobials and had ≥1 documented site-specific infection. The antimicrobial regimen prescribed by the ward physician (WARD regimen), recommended by the DigitalAMS™ (APP regimen), and recommended by two independent infectious disease (ID) physicians before (Emp-ID regimen) and after (Def-ID regimen) the final microbiological results became available were compared in a pairwise fashion. The percent agreement of antimicrobial prescribing between the APP and Emp-ID regimens was 85.7% in the bacteremia group, 59.1% in the pneumonia group, 78.6% in the UTI group, and 85.2% in the SSTI group. The percent agreement between the APP and Emp-ID regimens was significantly higher than that between the WARD and Emp-ID regimens in three site-specific infection groups: the bacteremia group (85.7% vs. 47.9%, *p* < 0.001), the UTI group (78.6% vs. 37.8%, *p* < 0.001), and the SSTI group (85.2% vs. 40.2%, *p* < 0.001). Furthermore, the percent agreement between the APP and Def-ID regimens was similar to that between the Emp-ID and Def-ID regimens in all sites of infection. In conclusions, the implementation of DigitalAMS™ seems useful but needs some revisions. The dissemination of this ready-to-use application with customized clinical practice guidelines to other hospital settings may be beneficial.

## 1. Introduction

The emergence of antimicrobial resistance is a growing problem worldwide [1,2]. Antimicrobial resistance results in higher mortality and morbidity, a longer hospital length of stay, and increased healthcare costs [3]. Many previous studies have revealed an association between the emergence of antimicrobial resistance and antimicrobial exposure [4,5].

Appropriate antimicrobial therapy is considered a key component in preventing and containing the emergence of antimicrobial resistance [6]. Many studies have confirmed that various antimicrobial stewardship (AMS) strategies, such as implementing clinical practice guidelines (CPGs), using computerized clinical decision support programs, providing individual feedback, and performing bedside infectious disease (ID) consultations are effective in reducing inappropriate antimicrobial therapy [6,7,8,9,10,11,12]. Although developing CPGs is not a sophisticated task, implementing and maintaining CPG compliance can be challenging [12,13,14].

The 2016 Infectious Diseases Society of America and Society for Healthcare Epidemiology of America guidelines for implementing an AMS program recommend the development of facility-specific CPGs accompanied by a dissemination and implementation strategy [15]. The guidelines also suggest incorporating computerized clinical decision support at the time of prescribing antimicrobials into the AMS program [15].

In 2018, the Division of Infectious Diseases and Tropical Medicine, Faculty of Medicine Siriraj Hospital, Mahidol University, developed and implemented the DigitalAMS™ application. DigitalAMS™ is a ready-to-use offline mobile-based application for Siriraj Hospital healthcare personnel. This application is used to choose an empirical antimicrobial regimen for four common site-specific infections based on the existing CPGs for relevant healthcare personnel at Siriraj Hospital. In addition to the hospital CPG, some AMS interventions such as education programs, customized antibiograms, drug use evaluation programs, and antibiotic authorization were successfully implemented at our hospital before the DigitalAMS™ project.

We conducted the present cross-sectional study to evaluate the use of the DigitalAMS™ application. Our primary objective was to determine the results of the implementation of DigitalAMS™ in choosing empirical antimicrobial therapy for four common site-specific infections (bacteremia, pneumonia, urinary tract infection (UTI), and skin and soft tissue infection (SSTI)) by using the ID physician’s decision as the gold standard. Our secondary objective was to determine the limitations of using the application and accordingly improve its implementation.

## 2. Methods

### 2.1. Study Design and Population

We conducted a cross-sectional study at Siriraj Hospital from January to June 2018. Siriraj Hospital is a 2200-bed university hospital located in Bangkok, Thailand. The study was approved by the Siriraj Institutional Review Board, and the requirement for informed consent was waived (certification of approval number: Si 700/2017).

Hospitalized patients aged ≥18 years who met the following inclusion criteria were eligible for participation: (1) They had received at least one dose of an antimicrobial agent during hospitalization, and (2) they had at least one of the following site-specific infections: bacteremia, pneumonia, UTI, or SSTI. We used Centers for Disease Control and Prevention/National Healthcare Safety Network surveillance definitions for bacteremia, pneumonia, UTI, and SSTI [16]. If an eligible patient had more than one site of infection within the same period, each site of infection was assessed separately. In patients with multiple sites of infection, it was sometimes difficult to distinguish which antibiotic was prescribed for treatment at which site. Therefore, all antibiotics prescribed during such infection episodes were considered for treatment of the index infection. If an eligible patient had more than one episode of infection during hospitalization, only the first episode of infection was included.

### 2.2. Development of Hospital CPGs and DigitalAMS™ Application

DigitalAMS™ is an offline mobile application specifically designed to help relevant healthcare personnel choose appropriate empirical antimicrobial regimens. A user needs to enter only three important patient-related factors: (1) the patient’s risk stratification (admission to the general ward or intensive care unit), (2) the risk of acquisition of multidrug-resistant pathogens, and (3) the suspected site of infection. After the user has entered these three factors, the application will suggest the first-line regimen and an alternative regimen of empiric antimicrobial therapy. To access the application, a first-time user must register with a designated username and an individual password before downloading the application via the internet. Once the application has been successfully downloaded and installed on the user’s mobile phone, the user can access the application anywhere without an internet connection. The application is a health initiative owned, operated, and provided by MSD Pharmaceuticals Private Limited (MSD), although MSD does not have liability relating to the output of the application. The application is provided to Siriraj Hospital at no cost. Figure 1 shows the interface of the DigitalAMS™ application.

The Siriraj CPGs of the empirical antimicrobial regimen for each targeted site of infection were customized based on the pharmacokinetic/pharmacodynamics of the antibiotics, the hospital antibiogram, and previous studies conducted at Siriraj Hospital [7,17,18]. The antimicrobial regimens were finally approved by the Division of Infectious Diseases and Tropical Medicine, Department of Medicine, Faculty of Medicine Siriraj Hospital. Appendix A shows the details of the antimicrobial regimens according to the patients’ risk stratification, risk of acquisition of multidrug-resistant pathogens, suspected site of infection, and all related definitions. These hospital CPGs were distributed in a published version to all clinical departments before the implementation of the application. Additionally, the online version of the CPGs was available to all hospital healthcare personnel.

### 2.3. Data Collection

Potential participants in this study were identified through the hospital’s electronic database. A medical record review was performed on all hospitalizations with at least one International Classification of Disease (ICD) code of the index infection and with at least one prescription of an antimicrobial agent to determine the patient’s eligibility. Of all the potential participants, only those who met the study inclusion criteria were randomly enrolled by using a computer program.

An in-depth medical record review was subsequently performed to obtain all the necessary information, including demographics, baseline clinical characteristics, details of the index infection, laboratory and microbiological results, antimicrobial therapy, and treatment outcomes. Demographic data included age, sex, and weight (if available). Clinical data included a history of previous hospitalization, previous antimicrobial therapy, and underlying diseases. Details of the index infection were recorded in a narrative format and were accompanied by all necessary laboratory and microbiological results. Data on antimicrobial therapy, including type, dose, and duration, were also recorded.

Antimicrobial regimens were retrieved from four sources: (1) the antimicrobial regimen that the eligible patient received upon hospitalization (WARD regimen), (2) the antimicrobial regimen recommended by using the DigitalAMS™ application as empirical antimicrobial therapy (APP regimen), (3) the antimicrobial regimen recommended by the ID physician as empirical therapy before the microbiological results became available (Emp-ID regimen), and (4) the antimicrobial regimen recommended by the ID physician as the definitive therapy after the microbiological results became available (Def-ID regimen).

The WARD regimen was directly obtained by performing a medical record review, whereas the APP regimen was obtained by entering three important parameters into the DigitalAMS™ application. For the Emp-ID and the Def-ID regimens, two independent ID physicians reviewed the patient’s case record form as well as all the initial laboratory results with and without the final microbiological results. The ID physicians were masked to the WARD regimen and the patient’s treatment outcome. If disagreement occurred between the recommendations of two ID physicians, the recommendation of a third ID physician was considered final. The Def-ID regimen was available only among patients with identified causative pathogens.

### 2.4. Statistical Analysis

The agreement between each pair of recommendations was estimated to be approximately 50% with an allowable error of 10%, and the expected sample size was 97 patients per site-specific infection. After adjusting for a 10% loss to follow-up, the final sample size was 107 patients per site-specific infection or 438 patients for 4 site-specific infections.

An agreement of antimicrobial regimens was defined as the matching of two antimicrobial regimens (same drugs or different drugs with a similar spectrum). A too-broad or too-narrow antimicrobial regimen was considered mismatched. Examples of matching regimens include amoxicillin/clavulanic acid and ampicillin/sulbactam, ceftriaxone and cefotaxime, and meropenem and imipenem/cilastatin. Examples of mismatching regimens include ertapenem and meropenem, meropenem and colistin, and vancomycin and linezolid.

Categorical variables are reported as frequency and percentage, and continuous variables are shown as mean ± standard deviation. We calculated the percent agreement by performing a pairwise comparison of each site-specific infection: (1) the WARD regimen and APP regimen, (2) the APP regimen and Emp-ID regimen, (3) the APP regimen and Def-ID regimen, and (4) the WARD regimen and Emp-ID regimen. The percent agreement of the APP regimen and Emp-ID regimen was considered to be the concordance of the application, which was the primary outcome. The percent agreement was reported with its 95% confidence interval. All analyses were performed with Stata/IC version 14.0 (StataCorp, College Station, TX, USA).

## 3. Results

During the 6-month study period, there were 7995 hospitalizations with at least 1 antimicrobial prescription and at least 1 ICD-10 code of target site-specific infections. Of these 7995 hospitalizations, we performed a medical record review and randomly enrolled a total of 401 episodes of the index infection (314 unique patients): 98 patients with bacteremia, 98 with pneumonia, 103 with UTI, and 102 with SSTI. The study flowchart is shown in Figure 2.

Table 1 shows the patients’ baseline characteristics, clinical characteristics, and treatment outcomes stratified by the site of infection. The mean age of patients with bacteremia, pneumonia, UTI, and SSTI was 69.05 ± 15.20, 72.35 ± 16.82, 71.81 ± 14.42, and 62.59 ± 19.90 years, respectively (*p* = 0.006). Underlying diseases were comparable between the four groups. However, the proportion of cerebrovascular disease was slightly higher in the pneumonia group (37.7%) and the UTI group (37.8%) when compared with the bacteremia group (27.5%) and the SSTI group (16.6%, *p* = 0.002). Furthermore, the SSTI group was more likely to have community-acquired infections (50.0%), and the UTI group was more likely to have hospital-acquired infections (61.1%, *p* < 0.001) than the other groups. The proportion of ventilator dependency was highest in the pneumonia group (36.7%, *p* < 0.001).

The leading causative pathogen was *Escherichia coli* in the bacteremia group (53.0%), *Pseudomonas aeruginosa* in the pneumonia group (12.2%), *E. coli* in the UTI group (40.7%), and methicillin-susceptible *Staphylococcus aureus* in the SSTI group (10.7%). Piperacillin/tazobactam was the most commonly prescribed antimicrobial agent in the bacteremia group (46.9%), the pneumonia group (63.2%), and the UTI group (53.4%), whereas amoxicillin/clavulanic acid was the most commonly prescribed antimicrobial agent in the SSTI group (38.2%).

With respect to treatment outcomes, the 28-day mortality rate was highest in the pneumonia group (30.6%), followed by the bacteremia group (22.4%), the UTI group (12.6%), and the SSTI group (6.8%) (*p* < 0.001). The 28-day infection-related mortality rate, the overall hospital mortality rate, and the length of stay were also highest in the pneumonia group.

Table 2 shows the percent agreement of each comparison group stratified by the infection site. The percent agreement between the APP regimen and the Emp-ID regimen was 85.7% in the bacteremia group, 59.1% in the pneumonia group, 78.6% in the UTI group, and 85.2% in the SSTI group. The percent agreement between the WARD regimen and the Emp-ID regimen was only 47.9% in the bacteremia group, 55.1% in the pneumonia group, 37.8% in the UTI group, and 40.2% in the SSTI group. The percent agreement between the APP regimen and the Emp-ID regimen was significantly higher than that between the WARD regimen and the Emp-ID regimen in three site-specific infection groups: the bacteremia group (85.7% vs. 47.9%, *p* < 0.001), the UTI group (78.6% vs. 37.8%, *p* < 0.001), and the SSTI group (85.2% vs. 40.2%, *p* < 0.001). The percent agreement between the APP regimen and the Def-ID regimen was 54.1% in the bacteremia group, 50.0% in the pneumonia group, 46.2% in the UTI group, and 64.6% in the SSTI group. The percent agreement between the Emp-ID regimen and the Def-ID regimen was 55.1% in the bacteremia group, 68.2% in the pneumonia group, 42.9% in the UTI group, and 68.8% in the SSTI group. The percent agreement between the APP regimen and the Def-ID regimen and that between the Emp-ID regimen and the Def-ID regimen was comparable at all sites of infection (all *p* > 0.05).

## 4. Discussion

Our study results revealed that DigitalAMS™ provided an acceptable percent agreement in nearly all the sites of the index infection except for pneumonia when using the ID physician’s recommendation as the gold standard. Although the percent agreement between the APP regimen and the Emp-ID regimen did not reach 100%, the percent agreement was significantly higher than that of the WARD regimen and the Emp-ID regimen. Additionally, the percent agreement between the APP regimen and the Def-ID regimen was similar to that between the Emp-ID regimen and the Def-ID regimen at all sites of infections. This suggests that the APP regimen may provide benefits comparable to those obtained by an ID specialist’s recommendation.

We also investigated the underlying causes of the lower percent agreement between the APP regimen and the Emp-ID regimen within the pneumonia group (59.1%). The disagreement commonly occurred among patients with a history of aspiration and those who may require an anti-anaerobic agent. Given that the 2018 version of the hospital’s CPG did not include a regimen for aspiration pneumonia, more than one-third of the APP regimen did not match the Emp-ID regimen. These findings resulted in a subsequent revision of the hospital’s CPGs.

The results of our study are similar to those of previous studies [17,19]. The DigitalAMS™ application helps to improve antimicrobial use and thus optimize patient care. Priorities can be identified based on the patient location (general ward vs. intensive care unit) and the antimicrobial spectrum, antimicrobial cost, duration of therapy, and the number of different antimicrobials prescribed to a single patient. Such mobile applications must be designed and tested by members of the AMS program, and new applications and updates are needed.

Our study had two main strengths. First, to our knowledge, this is the first study in Thailand that was specifically designed to evaluate the implementation of the DigitalAMS™ application in choosing empirical antimicrobial therapy for four common site-specific infections. Second, in addition to using the ID physician’s recommendation as the gold standard, we also obtained a recommendation for definitive antimicrobial therapy from DigitalAMS™ for patients with identified causative pathogens.

Our study also had three main limitations. First, the retrospective nature of the study may have resulted in information bias. The antimicrobial regimen recommended by the DigitalAMS™ and the antimicrobial regimen recommended by the ID physician relied on information from the medical record review. Therefore, some information may have been missing, negatively influencing the ID physician’s decision. Second, the study did not have enough power to determine the association between the percent agreement of the WARD and APP regimens and the treatment outcomes because this study was designed to determine the agreement of the APP and the Emp-ID regimens as the primary outcome, and the sample size was computed to detect the primary outcome. Third, this study was conducted at a university hospital in Thailand. Therefore, its results may not be applicable to other hospital settings.

In conclusion, the implementation of DigitalAMS™ for choosing an appropriate antimicrobial regimen for empirical treatment of four common site-specific infections in hospitalized patients at Siriraj Hospital seems to be useful. However, both this mobile application and the hospital CPGs need some revisions to improve the concordance between them. The dissemination of this ready-to-use application with the customized CPGs to other hospital settings may be beneficial. A future multi-center study is necessary to fully explore the benefits of the application.

## Figures and Tables

**Figure 1 antibiotics-12-00113-f001:**
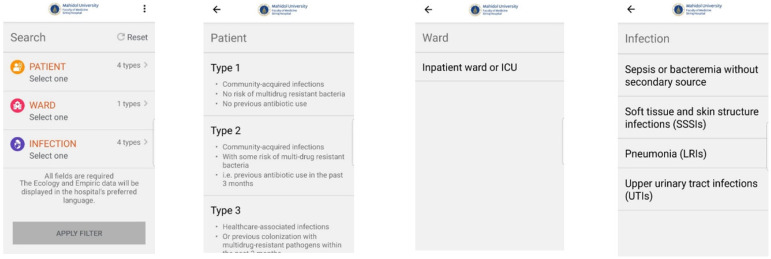
The DigitalAMS^TM^ application interfaces.

**Figure 2 antibiotics-12-00113-f002:**
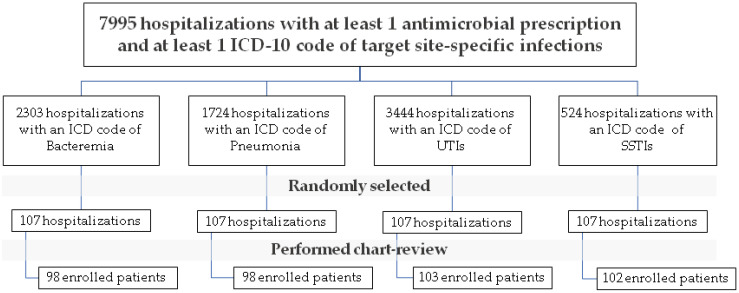
Study flowchart.

**Table 1 antibiotics-12-00113-t001:** Baseline characteristics, clinical features, and treatment outcomes, stratified by the site-specific infection.

Variables	Bacteremia (*n* = 98)	Pneumonia (*n* = 98)	UTI (*n* = 103)	SSSI (*n* = 102)	*p*-Value
**Baseline Characteristics**					
Male	42 (40.8%)	59 (60.2%)	44 (42.7%)	52 (50.9%)	0.04
Mean age (+/− SD), year	69.05 ± 15.20	72.35 ± 16.82	71.81 ± 14.42	62.59 ± 19.90	0.006
Previous hospitalizations in the past 3 months	12 (12.2%)	7 (7%)	9 (8%)	15 (14.7%)	0.37
Underlying disease					
− Hypertension	66 (67.3%)	58 (59.1%)	69 (66.9%)	63 (61.7%)	0.56
− Cerebrovascular disease	27 (27.5%)	37 (37.7%)	39 (37.8%)	17 (16.6%)	0.002
− Chronic kidney disease	26 (26.5%)	14 (14.2%)	24 (23.3%)	24 (23.5%)	0.19
− Cardiovascular disease	36 (36.7%)	26 (26.5%)	34 (33%)	22 (21.5%)	0.07
− Diabetes mellitus	48 (48.9%)	32 (32.6%)	42 (40.7%)	42 (41.1%)	0.14
− Hematologic disease	11 (11.2%)	12 (12.2%)	7 (6.8%)	6 (5.8%)	0.30
− Non-hematologic malignancy	28 (28.5%)	22 (22.4%)	30 (29.1%)	30 (29.4%)	0.65
− Solid malignancy	28 (28.5%)	23 (23.4%)	26 (25.2%)	16 (15.6%)	0.17
− Organ transplant	1 (1%)	1 (1%)	0 (0%)	0 (0%)	0.37
− Immunosuppressant agents in the preceding 30 days	18.3% (18)	16.3% (16)	18.4% (18)	14.7% (15)	0.91
− Neutropenia	6 (6.1%)	3 (3%)	2 (1.9%)	2 (1.9%)	0.32
− HIV infection	1 (1%)	1 (1%)	2 (1.9%)	0 (0%)	0.58
Clinical features					
Type of infections					
− Community-acquired infection	34 (34.6%)	45 (45.9%)	29 (28.1%)	51 (50%)	0.001
− Community-acquired infection with risk for MDR pathogen	7 (7.1%)	9 (9.1%)	11 (10.6%)	17 (16.6%)	
− Hospital-acquired infection	56 (57.1%)	44 (44.9%)	63 (61.1%)	33 (32.3%)	
− Hospital-acquired infection, while receiving carbapenem therapy	1 (1%)	0 (0%)	0 (0%)	1 (1%)	
Severity of illness					
Ventilator dependency	15 (15.3%)	36 (36.7%)	15 (14.5%)	7 (6.8%)	<0.001
Acute respiratory distress syndrome	2 (2%)	3 (3%)	1 (0.9%)	0 (0%)	0.23
Causative pathogen (s)					
Gram-positive bacteria	19 (19.3%)	14 (14.2%)	13 (12.6%)	20 (19.6%)	0.43
− MSSA	2 (2%)	12 (12.2%)	2 (1.9%)	11 (10.7%)	0.002
− MRSA	0 (0%)	0 (0%)	0 (0%)	0 (0%)	NA
− *Streptococcus pneumoniae*	8 (8.1%)	2 (2%)	1 (0.9%)	6 (5.8%)	0.04
− *Enterococcus faecalis*	7 (7.1%)	0 (0%)	9 (8.7%)	4 (3.9%)	0.01
− *Enterococcus faecium*	0 (0%)	0 (0%)	1 (0.9%)	3 (2.9%)	0.18
Gram-negative bacteria	73 (74.4%)	30 (30.6%)	68 (66%)	25 (24.5%)	<0.001
− *Escherichia coli*	52 (53%)	4 (4%)	42 (40.7%)	6 (5.8%)	<0.001
Ceftriaxone-resistant *E. coli*	30 (30.6%)	3 (3%)	28 (27.1%)	0 (0%)	<0.001
− *Klebsiella pneumoniae*	10 (10.2%)	10 (10.2%)	8 (7.7%)	2 (1.9%)	0.05
Ceftriaxone-resistant *K. pneumoniae*	6 (6.1%)	4 (4%)	6 (5.8%)	0 (0%)	0.04
− *Pseudomonas aeruginosa*	7 (7.1%)	12 (12.2%)	11 (10.6%)	9 (8.8%)	0.65
Multi-drug resistant *P. aeruginosa **	1 (1%)	1 (1%)	5 (4.8%)	0 (0%)	0.07
− *Acinetobacter baumannii*	4 (4%)	6 (6.1%)	1 (0.9%)	0 (0%)	0.02
Multi-drug resistant *A. baumannii **	2 (2%)	3 (3%)	1 (0.9%)	0 (0%)	0.23
Empirical antimicrobial regimen (WARD regimen)					
Penicillin	9 (9.1%)	3 (3%)	4 (3.8%)	6 (5.8%)	0.27
Cephalosporins	50 (51%)	31 (31.6%)	46 (44.6%)	53 (51.9%)	0.02
− Ceftriaxone	43 (43.8%)	20 (20.4%)	40 (38.8%)	24 (23.5%)	<0.001
Amoxicillin/clavulanic acid	7 (7.1%)	32 (32.6%)	12 (11.6%)	39 (38.2%)	<0.001
Piperacillin/tazobactam	46 (46.9%)	62 (63.2%)	55 (53.4%)	28 (25.4%)	<0.001
Meropenem	39 (39.8%)	18 (18.3%)	28 (27.1%)	13 (12.7%)	<0.001
Fluoroquinolones	13 (13.2%)	26 (26.5%)	16 (15.5%)	28 (27.5%)	0.02
Macrolide	3 (3%)	20 (20.4%)	3 (2.9%)	3 (2.9%)	<0.001
Vancomycin	11 911.2%)	6.1% (6)	6 (5.8%)	11 (10.7%)	0.35
Aminoglycoside	8 (8.1%)	1 (1%)	5 (4.8%)	5 (4.9%)	0.14
Clindamycin	1 (1%)	5 (5.1%)	2 (1.9%)	20 (19.6%)	<0.001
Metronidazole	4 (4%)	2 (2%)	2 (1.9%)	3 (2.9%)	0.77
Treatment outcomes					
Favorable clinical outcome	77 (78.6%)	67 (68.4%)	90 (87.4%)	94 (92.2%)	<0.001
Favorable microbiological outcome	52 (53%)	10 (10.2%)	38 (36.9%)	16 (15.7%)	<0.001
28-day over mortality	22 (22.4%)	30 (30.6%)	13 (12.6%)	7 (6.8%)	<0.001
28-day ID-related mortality	20 (20.4%)	23 (23.7%)	11 (10.6%)	7 (6.8%)	<0.001
Status at discharge					
− Survivor	74 (75.5%)	64 (65.3%)	88 (85.4%)	93 (91.1%)	<0.001
− Non-survivor	22 (22.4%)	33 (33.3%)	13 (12.6%)	7 (6.8%)	
− Against advice	2 (2%)	1 (1%)	2 (2%)	2 (2%)	
Length of hospital stay, days	18.62 ± 14.47	16.45 ± 18.41	14.40 ± 10.86	13.99 ± 13.89	<0.001

Abbreviations: UTI, urinary tract infection; SSTI, skin and soft tissue infection; SD, standard deviation; HIV, human immunodeficiency virus; MDR, multi-drug resistant; MSSA, methicillin-susceptible *Staphylococcus aureus*; MRSA, methicillin-resistant *Staphylococcus aureus*; ID, infectious disease. * Multidrug resistance is defined as resistance to at least one antimicrobial agent from at least three classes.

**Table 2 antibiotics-12-00113-t002:** Percent agreement of each comparison group, stratified by the infection site.

Variables	Comparator-1	Comparator-2	Percent Agreement
Bacteremia (*n* = 98)		Pneumonia (*n* = 98)	UTI (*n* = 103)		SSTI (*n* = 102)	
Primary outcome	APP	Emp-ID	84 (85.7%)	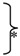	58 (59.1%)	81 (78.6%)	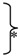	87 (85.2%)	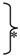
Secondary outcomes	WARD	Emp-ID	47 (47.9%)	54 (55.1%)	39 (37.8%)	41 (40.2%)
WARD	APP	48 (48.9%)		33 (33.6%)	35.9 (37%)		38 (37.2%)	
APP	Def-ID **	53/98 (54.1%)		22/44 (50.0%)	42/91 (46.2%)		31/48 (64.6%)	
Emp-ID	Def-ID **	54/98 (55.1%)		30/44 (68.2%)	39/91 (42.9%)		33/48 (68.8%)	

* *p*-value from the pairwise comparison <0.001. ** The causative pathogens were not identified in some study subjects. UTI, urinary tract infection; SSTI, skin and soft tissue infection.

## Data Availability

The study dataset is available from the corresponding author upon reasonable request.

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
