# Peer review of "Development and Implementation of a Mobile Application for Choosing Empirical Antimicrobial Therapy for Bacteremia, Pneumonia, Urinary Tract Infection, and Skin and Soft Tissue Infection among Hospitalized Patients"

_antibiotics, 2023, doi:10.3390/antibiotics12010113_

Round 1
Reviewer 1 Report
Dear authors,
Thanks for this study - thanks for all your hard work. As we urgently need applications that could help with rational abx prescribing, this is a very relevant study.
In order to interpret the results, I need some clarification regarding the following: 401/7995 admissions that were eligible were included, via a randomization process. Could you please clarify - beyond lines 116-119- details of the randomization process. What sample size were you aiming for? Furthermore- a flow diagram of participant selection would be helpful.
Furthermore, a shortcoming of this study is the mono-center aspect. Could you elaborate on this in the discussion section?
Reviewer 2 Report
Dear authors,
Congratulations for the research.
The study set out the performance of a digital strategy to assist in the adherence of CPGs. In fact, these devices, when properly incorporated based on local hospital epidemiology, can reach a greater number of users and contribute to the dissemination of institution’s knowledge and the practice of correct antibiotic prescription.
In this sense, the article was well structured and properly answer its first objective: the percentage agreement of the APP regimen and Emp-ID regimen.
It is also a very useful and appropriate research that would be published to demonstrate how good strategies might be implemented on the hospital practice to increment antimicrobial stewardship.
However, it might be necessary some considerations for the authors:
Tabel 1. For Gram-negative bacteria, there is twice of ceftriaxone -resistance strain and twice of multi-drug resistant strain. I think the authors need to add each one, because it related the same data stratified by the site-specific infection.
Table 1. Multi-drug resistant strain: is an item of Gram-negative bacteria, but I would like to know which the meaning that the authors assume as MDR in this study. In my institution, for example, for GNB we considered all Enterobacterales resistant to carbapenems (CRE). So, it would be nice to put as a legend in the bottom of the table 1.
Tabel 1. Page 6, please put the information of variables and the site-specific infection as a line, because it was really difficult for the reader to follow the information when there was a break of the table 1 on page 6. And, probably because of this there were some mistakes related with the outcomes. See on lines: 187, 188,189 and 190. The 28-day mortality rate and others outcomes related to treatment that authors point out, was highest in the bacteremia group. Please it would be necessary to rewrite this paragraph.
Round 2
Reviewer 1 Report
Thanks for the edits - the current version is fine for me. Thanks for all the hard work!